# Genes Implicated in Familial Parkinson’s Disease Provide a Dual Picture of Nigral Dopaminergic Neurodegeneration with Mitochondria Taking Center Stage

**DOI:** 10.3390/ijms22094643

**Published:** 2021-04-28

**Authors:** Rafael Franco, Rafael Rivas-Santisteban, Gemma Navarro, Annalisa Pinna, Irene Reyes-Resina

**Affiliations:** 1Department Biochemistry and Molecular Biomedicine, University of Barcelona, 08028 Barcelona, Spain; rfranco@ub.edu (R.F.); rrivasbioq@gmail.com (R.R.-S.); ire-reyes@hotmail.com (I.R.-R.); 2Centro de Investigación Biomédica en Red Enfermedades Neurodegenerativas (CiberNed), Instituto de Salud Carlos III, 28031 Madrid, Spain; g.navarro@ub.edu; 3Department Biochemistry and Physiology, School of Pharmacy and Food Sciences, University of Barcelona, 08028 Barcelona, Spain; 4National Research Council of Italy (CNR), Neuroscience Institute–Cagliari, Cittadella Universitaria, Blocco A, SP 8, Km 0.700, 09042 Monserrato (CA), Italy

**Keywords:** mitochondria, vesicular transport, mitophagy, Lewy bodies, synuclein aggregation, familial Parkinson’s disease, early-onset Parkinson’s disease

## Abstract

The mechanism of nigral dopaminergic neuronal degeneration in Parkinson’s disease (PD) is unknown. One of the pathological characteristics of the disease is the deposition of α-synuclein (α-syn) that occurs in the brain from both familial and sporadic PD patients. This paper constitutes a narrative review that takes advantage of information related to genes (*SNCA, LRRK2, GBA, UCHL1, VPS35, PRKN, PINK1, ATP13A2, PLA2G6, DNAJC6, SYNJ1, DJ-1/PARK7* and *FBXO7*) involved in familial cases of Parkinson’s disease (PD) to explore their usefulness in deciphering the origin of dopaminergic denervation in many types of PD. Direct or functional interactions between genes or gene products are evaluated using the *Search Tool for the Retrieval of Interacting Genes/Proteins* (STRING) database. The rationale is to propose a map of the interactions between *SNCA*, the gene encoding for α-syn that aggregates in PD, and other genes, the mutations of which lead to early-onset PD. The map contrasts with the findings obtained using animal models that are the knockout of one of those genes or that express the mutated human gene. From combining in silico data from STRING-based assays with in vitro and in vivo data in transgenic animals, two likely mechanisms appeared: (i) the processing of native α-syn is altered due to the mutation of genes involved in vesicular trafficking and protein processing, or (ii) α-syn mutants alter the mechanisms necessary for the correct vesicular trafficking and protein processing. Mitochondria are a common denominator since both mechanisms require extra energy production, and the energy for the survival of neurons is obtained mainly from the complete oxidation of glucose. Dopamine itself can result in an additional burden to the mitochondria of dopaminergic neurons because its handling produces free radicals. Drugs acting on G protein-coupled receptors (GPCRs) in the mitochondria of neurons may hopefully end up targeting those receptors to reduce oxidative burden and increase mitochondrial performance. In summary, the analysis of the data of genes related to familial PD provides relevant information on the etiology of sporadic cases and might suggest new therapeutic approaches.

## 1. Introduction

Parkinson’s disease is a prevalent neurodegenerative disease the first clinical symptoms of which include tremors and motor dysfunction. The number of cases with unknown causes is much higher than the number of familial cases; that is, cases due to known genetic alterations. Years ago, the percentage of cases of inheritable PD was estimated to be around 5%. Recent estimates increased this percentage because of Genetic-oriented research, which led to the discovery of more genes associated with the disease. In a recent multicenter study in a cohort of 1587 cases, mutations were found in 14.1% of patients [1] (see also [2,3]). Although familial cases often display early-onset symptoms, the main risk factor is age with 60 years being considered as the threshold for developing symptoms in sporadic cases. In these cases, many positive (e.g., caffeine) and negative (e.g., toxin exposure) risk factors are known [4,5,6,7,8,9,10,11,12,13], but the exact causes of why dopamine neurons of the substantia nigra die in both familial and sporadic cases remain unknown.

The work of Hornykiewicz and colleagues was instrumental for early therapeutic intervention in PD cases. They discovered the lack of dopamine in certain brain areas as the cause of motor symptoms and noticed that dopamine supplementation was not effective because the compound is unable to cross the blood–brain barrier. They discovered that L-DOPA (levodopa), the precursor of the neurotransmitter, was able to enter the brain and there be converted into dopamine. L-DOPA is still used in PD therapy. The need for chronic treatment and fluctuations in drug levels in the brain may lead to some side effects, mainly dyskinesia [14,15,16,17,18,19]. Side effects may be addressed via surgical procedures. Since the nineties, the technique has been refined and used with success by implanting electrodes that achieve what is known as deep-brain stimulation (DBS) [20,21,22,23,24,25,26,27,28]. Unfortunately, there is no therapeutic intervention that delays disease progression, that is, the neurodegeneration of dopaminergic nigral neurons.

PD results from death of dopamine-producing neurons of the substantia nigra. An imbalance of dopaminergic neurotransmission in this area of the brain leads to motor deficits, which first appear as tremors and difficulty starting to walk and moving limbs precisely. Be it cause or consequence, there is a main characteristic of pathological PD: the appearance of Lewy bodies formed by aggregates of a protein, α-synuclein (α-syn), the function of which has not yet been fully elucidated. PD has analogies and differences with other diseases known as α-synucleinopathies; the two other main α-synucleinopathies are dementia with Lewy bodies and multiple system atrophy (see [29,30,31]). The deposition of α-syn aggregates occurs in the brain of patients with both familial and sporadic PD. Finally, it should be noted that a theory has arisen that postulates that the risk of PD may be increased by some viral infections [32].

Taking into account the familial cases, i.e., mutations in *SNCA, LRRK2, GBA, UCHL1, VPS35, PRKN, PINK1, ATP13A2, PLA2G6, DNAJC6, SYNJ1, DJ-1/PARK7* and *FBXO7* genes, it appears that (i) mutant forms of α-syn can aggregate and lead to early-onset PD and (ii) that mutations in other genes lead to deposition of non-mutated α-syn and early onset of the EP. The present narrative review aims to propose a scenario in which these players are connected; that is, we would like to obtain information from familial cases that could help decipher the causes of dopaminergic denervation in all types of PD.

The approach is sustained in two qualitatively different datasets. First, a blind approach is used to find the connections between genes that have alterations associated with the disease. First, we selected genes related to PD familial cases (listed in Table 1) to find interactions among them (in a blind-like approach) using the *Search Tool for the Retrieval of Interacting Genes/Proteins* (STRING). Second, information from transgenic models was analyzed in the search for mechanisms of phenotypic alterations (focused on the CNS and, whenever possible, considering alterations in the substantia nigra/striatum). From these two independent sets of data we finally presented a picture of the potential events that occur in degenerating dopaminergic neurons.

## 2. STRING Analysis Restricted to Connections Obtained from Experimental, Gene Fusion, Protein Analogy or Proximity Data

STRING 11.0 version was used in the analysis of the potential connections among genes in Table 1. Both software and platform are incorporated into the European “ELIXIR Core Data Resources”; it is freely available and analyses were performed on-line in https://string-db.org (accessed on 4 March 2021).

The STRING default setting for the genes associated with familial PD and listed in Table 1 leads to Figure 1A, in which virtually all genes (or gene products) are interconnected. We have omitted genes that were more recently associated with familial PD but for which less data was available [33,34]. Figure 1A shows that *PLA2G6* is linked to *UCHL1* and *BNAJc6* only by neighborhood, meaning that there are no studies confirming whether or not there is any functional link between them. Obviously, at the bibliographic level, all genes are related to Parkinson’s disease; therefore, we next eliminated the connections derived from the mention of genes in papers related to PD.

At first glance, avoiding the bias due to appearance in PD-related papers, the number of interactions is markedly reduced. On the one hand, restricting connections to those derived from experimental data, gene fusion, protein analogy, or proximity, 5 gene products were found to be interconnected (See Figure 1B). All except the *PARK2/PARK7* potential interaction were (according to STRING database) experimentally determined (pink line between elements). On the other hand, an interaction between *SYNJ1* and *DNAJC6* products was identified. Overall, 6 elements were not connected to each other. If we enrich the pattern, including partners that were not present in the initial analysis, we obtain Figure 1C, in which the 6 elements still remain non-connected. Patterns arising using these 6 genes as individual elements are in the Appendix A (settings exclude co-appearance in PD-related papers). Whereas 5 of them showed interrelationships with other proteins, the product of *PLA2G6*, 85/88 kDa calcium-independent phospholipase A2, did not display any interaction related to experiments, gene fusion, co-expression, co-occurrence or neighborhood, meaning that its role is not yet well deciphered. In contrast, the protein encoded by *UCLH1*, ubiquitin carboxyl-terminal hydrolase isozyme L1, is an element with relevant links that include experimental data, protein homology and gene and protein coexpression. Actually, the search of mechanisms for non-gene-associated PD cases is the aim of this paper.

## 3. Lessons from Interactions among *PARK2*, *PARK7*, *PINK1*, *LRRK2* and *VPS35*

The proteins encoded by *PARK2, PARK7, PINK1, LRRK2* and *VPS35* have a complementary role related to subcellular organelles, mainly the lysosome. Furthermore, practically all gene products participate in protein processing, in which ubiquitination is a relevant mechanism. Ubiquitination was described decades ago as a mechanism for targeting a protein for degradation [35,36,37]. If the processes involving ubiquitination are not completely balanced, the handling of proteins is inadequate, and the cell is damaged. Apart from ubiquitin-protein ligase (*PARK2*/parkin), other elements also participate in ubiquitination-mediated processes that lead to proteasomal degradation. This view is in agreement with the finding that some inherited PD cases are associated with the PARK7, PARK6, and PARK2 loci [38]. Mutant forms of *DJ-1/PARK7* that are associated with PD lead to differential interactions with E3 ubiquitin-protein ligase leading to altering protein processing or leading to oxidative stress [39]. *PINK1* is genetically associated with parkin. The discovery first made in *Drosophila melanogaster* [40,41] was later confirmed in humans [42]. In 2005, Smith and colleagues demonstrated an interaction of leucin-rich repeat kinase 2 (product of *LRRK2*) with parkin and showed that mutant forms of the kinase may induce neuronal death [43]. Vacuolar protein sorting-associated protein 35 is involved in the vesicular transport of vesicles from mitochondria. Such mitochondrial-derived vesicles are linked to ubiquitination in a complex way (see [44] for details). Furthermore, the ubiquitin carboxyl-terminal hydrolase isozyme L1 (*UCHL1*) and the F-box component of protein 3 ubiquitin-protein ligase complex (*FBXO7*), have a role in targeting protein degradation using the ubiquitination pathway.

## 4. Lessons from Interactions between and from SYNJ1 and DNAJC6

The link between these two genes is quite straightforward. On the one hand, *SYNJ1* codes for Synaptojanin-1, which is a quite unspecific phosphatase because it may degrade different phosphoinositides, which are key components on biological membranes. On the other hand, DNAJC6 codes for auxilin, which is a protein phosphatase involved in vesicle handling; in addition, auxilin belongs to a family of chaperones (DNAJ/HSP40). Knowledge of the function of these proteins is only partial, but at present they seem related to early events occurring shortly after endocytosis. Data on their involvement in the trafficking of intracellular vesicles or targeting of proteins for degradation is scant. In fact, genetic deletion of *DNAJC6* in mice results in death due to the inability to perform clathrin-mediated endocytosis [45]. A study in *Saccharomyces cerevisiae* confirms that both the product of *Dnajc6* and Synaptojanin-1 are necessary for the uncoating of clathrin, the proper trafficking of endosomes and the delivery of endocytic cargo [46,47].

## 5. Enriching the Network

STRING is a powerful tool as it permits the enrichment of networks but avoiding the bias of parkinsonism in the enrichment. In doing so, the *PARK2/PARK7/PINK1/LRRK2/VPS35* network connects with the *SYNJ1/DNAJC6* network. Obviously, this only is possible if more genes, related or not with PD, are considered. The resulting enriched network is in Figure 2.

First of all, Figure 2 shows that only 3 PD-related genes are not connected to the rest (*GBA, ATP132A* and *PLA2G6*). Importantly, *VPS35* becomes a relevant node that is connected, on the one side to several proteins involved in intracellular vesicular trafficking (vesicular sorting proteins, sorting nexins and Wnt Ligand Secretion Mediator, WLS) and, on the other side, directly to two gene products related to PD (*PINK1* and *LRRK2*) and, indirectly with all other PD-related genes except *GBA, ATP132A* and *PLA2G6*. Also of note is that the network involving most of the PD-related genes (Figure 2, see connections involving *VPS35*) contains elements having a variety of functions in common: ubiquitination, clathrin-mediated endocytosis and phosphoinositide handling. In the far bottom right of Figure 2, α-synuclein (α-syn) appears only connected to ubiquitin-60S ribosomal protein L40 due to protein homology.

In summary, the study of genes related with inherited forms of PD shows that the products of the genes participate in intracellular vesicular transport and in clathrin-mediated endocytosis, which is subsequent to the activation of cell surface G protein-coupled receptors (GPCRs).

## 6. Lessons from Transgenic Animals

Transgenic animals have been fundamental to understanding disease mechanisms. While in non-neurological diseases many models are based on the genetic inactivation of a gene in neurodegenerative diseases (providing so-called knockout (KO) animals), the approach of overexpressing a mutated or non-mutated form of a human protein is often used. In a set of tables, we present a summary of the data obtained from transgenic animals, both generated for the purpose of better understanding the disease of PD, or previously generated for another purpose, but having provided data on alterations of the functionality of the nervous system.

Data from transgenic animals of other PD-related genes and the relevant references are included in Table 3 and Table 4 and in Appendix A. After a selection of ad hoc papers, these 3 tables were constructed to summarize data from a significant number of studies indicating for each the transgenic model and the main anomalies found after in vivo or in vitro studies.

In accordance with the leading role of α-syn in synucleinopathies such as PD [48], there are several transgenic animals expressing different versions of the *SNCA* gene and several studies on them. The results of these studies are interesting despite their bias as the link between α-syn and synucleinopathies is obvious. On the one hand, the lack of a gene (mouse *SNCA* knockout) produced viable animals with some mild abnormalities. It is the overexpression of the human gene that led to neural alterations. On the other hand, the expression of a non-phosphorylatable (S129A) mutation on an *SNCA* knockout background did not lead to any significant alteration in neural function. Neural alterations including dopaminergic neuronal death appeared upon overexpression of some gene mutants: often the mutant version included a phosphorylatable amino acid or, the other way around, eliminates a phosphorylatable residue. Findings using such proteins that may be differentially phosphorylated could link α-syn to the PD-associated genes that code for a kinase or a phosphatase. Also of interest was fact that multimers of the protein in samples from a knockout model attenuate synaptic vesicle recycling and that the α-syn spread correlated with dopaminergic neurodegeneration. Apparently unrelated to PD-associated genes was an alteration in the effect of neurotrophic factors that were detected in some of those animals. Finally, we would like to highlight that some of the studies reported alterations in mitochondrial dynamics or function. If mutated α-syn has more or less phosphorylatable sites, this may be related to an energy imbalance; in fact, kinase action requires ATP.

Animals that are KO for the *Lrrk2* gene or overexpress human versions (mutant versions included) of this gene, consistently showed synaptogenetic deficiencies in intracellular vesicle trafficking and function and in protein processing. Interestingly, the G2019S mutation of the LRRK2 kinase produced neurodegeneration in a rodent model of human α-syn overexpression. The line expressing the G2019S mutation also displayed alterations in both homeostasis and neuronal morphogenesis.

**Table 3 ijms-22-04643-t003:** Findings in animal models related to *Snca*.

Animal Model (s)	In Vivo	In Vitro	Expression Level	Main Findings	Ref.
Mice harbouring *Snca^tm1Nbm^* mutation (n.s.n)		X	*Snca* KO mice	Modulates microglial activation phenotype Increased levels of reactive marker proteins	[49]
Mice harbouring *Snca^tm1Nbm^* mutation (n.s.n)		X	*Snca* KO mice	Altered palmitate metabolism	[50]
Mice harbouring *Snca^tm1Nbm^* mutation (n.s.n)		X	*Snca* KO mice	Mitochondrial lipid abnormality Electron transport chain impairment	[51]
Mice harbouring *Snca^tm1Nbm^* mutation (n.s.n)	X	X	*Snca* KO mice	Synaptic vesicle depletion	[52]
C57BL/6N-*Snca^tm1Mjff^*/J	X	X	*Snca* KO mice	Non-altered mitochondrial bioenergetics	[53]
B6; 129 × 1-*Snca^tm1Rosl^*/J		X	*Snca* KO mice	α-syn restricts RNA viral infections in the brain	[54]
B6; 129 × 1-*Snca^tm1Rosl^*/J		X	*Snca* KO mice	ROS and NOS-2 decreased in mature erythrocytes	[55]
B6; 129 × 1-*Snca^tm1Rosl^*/J		X	*Snca* KO mice	No modification in pale body-like inclusion Altered proteasome function	[56]
B6; 129 × 1-*Snca^tm1Rosl^*/J		X	*Snca KO* mice	Inhibition of intrasynaptic vesicle mobility Maintains recycling-pool homeostasis	[57]
B6; 129 × 1-*Snca^tm1Rosl^*/J	X		*Snca KO* mice	Cognitive impairments	[58]
B6; 129 × 1-*Snca^tm1Rosl^*/J		X	*Snca KO* mice	Vulnerability of peripheral catecholaminergic neurons to MPTP not regulated by α-synuclein	[59]
B6; 129 × 1-*Snca^tm1Rosl^*/J		X	*Snca KO* mice	Resistant to mitochondrial toxins	[60]
B6; 129 × 1-*Snca^tm1Rosl^*/J	X	X	*Snca KO* mice	Deficits in the nigrostriatal dopamine system	[61]
B6; 129 × 1-*Snca^tm1Rosl^*/J C57Bl/6JOlaHsd	X	X	*Snca* KO mice	Decreased impulsivity	[62]
B6; 129 × 1-*Snca^tm1Rosl^*/J C57Bl/6JOlaHsd	X		*Snca* KO mice	Neuromuscular pathology	[63]
B6; 129 × 1-*Snca^tm1Rosl^*/J C57Bl/6JOlaHsd	X	X	*Snca* KO mice	Decreased reuptake of dopamine in the dorsal striatum	[64]
B6; 129 × 1-*Snca^tm1Rosl^*/J (*Snca* KO) B6-^TgHSNCGtm1VLB^ (*Sncg* KO)	X	X	-α-syn (*Snca*) KO mice -ɣ-syn (*Sncg*) KO mice	Altered dopamine metabolism	[65]
B6; 129 × 1-*Snca^tm1Rosl^*/J (*Snca* KO) B6-^TgHSNCGtm1VLB^ (*Sncg* KO)	X	X	-α-syn (*Snca*) KO mice -ɣ-syn (*Sncg*) KO mice -α-ɣ-syn double KO mice	Increased striatal dopamine release Hyperdopaminergic signs	[66]
Triple-synuclein-KO (TKO)	X	X	α-syn (*Snca*), ß-syn (*Sncb*) and ɣ-syn (*Sncg*) triple KO mice	Functional alterations to the nigrostriatal system	[67]
Triple-synuclein-KO (TKO)	X	X	α-syn (*Snca*), ß-syn (*Sncb*) and ɣ-syn (*Sncg*) triple KO mice	Altered synaptic vesicle endocytosis α -, ß -, or γ-synucleins are functionally redundant	[68]
Triple-synuclein-KO (TKO)	X	X	α-syn (*Snca*), ß-syn (*Sncb*) and ɣ-syn (*Sncg*) triple KO mice	Age-dependent neuronal dysfunction	[69]
Triple-synuclein-KO (TKO) See paper for further details on model(s)	X	X	α-syn (*Snca*), ß-syn (*Sncb*) and ɣ-syn (*Sncg*) triple KO mice. Mice overexpressing human *SNCA* mutants PARK1/hA30P or PARK4/hα-syn	Effects on presynaptic architecture	[70]
B6;129 × 1-*Snca^tm1Rosl^*/J B6.CgTg (SNCA) OVX37Rwm *Snca^tm1Rosl^*/J	X	X	*-Snca* KO mice -*SNCA* overexpressing mice	Blockade of TrkB neurotrophic effect	[71]
B6;DBA-Tg (Thy1-SNCA) 61Ema		X	*SNCA* overexpressing mice	Impairment of mitochondrial function Elevated ROS in brain mitochondria	[72]
B6;DBA-Tg (Thy1-SNCA) 61Ema		X	*SNCA* overexpressing mice	Alterations in corticostriatal synaptic plasticity	[73]
B6;DBA-Tg (Thy1-SNCA) 61Ema	X	X	*SNCA* overexpressing mice	Alterations in calcium buffering capacity	[74]
(Thy1)-(WT)a-syn		X	*SNCA* overexpressing mice	Enhanced axonal degeneration after peripheral nerve lesion	[75]
B6.Cg-Tg (SNCA) OVX37Rwm *Snca^tm1Rosl^*/J	X	X	*SNCA* overexpressing mice	Deficits in dopaminergic transmission precede neuronal loss	[76]
B6.Cg-Tg (SNCA) OVX37Rwm *Snca^tm1Rosl^*/J		X	*SNCA* overexpressing mice	Impairment of macroautophagy in dopaminergic neurons	[77]
B6; 129 × 1-*Snca^tm1Rosl^*/J neurons transfected with *SNCA* or TsixK-*SNCA*		X	Cultured neurons from *Snca* KO mice overexpressing *SNCA* or TsixK-*SNCA*	α-syn multimers attenuate synaptic vesicle recycling	[78]
S129 mutations performed on mice harbouring Sncatm1Nbm mutation (n.s.n.)	X	X	S129D-*SNCA* (phosphomimetic) or S129A- *SNCA* (non-phosphorylatable) overexpressing *Snca* KO mice	No abnormalities detected (*in vivo*)	[79]
FVB;129S6-*Snca^tm1Nbm^* Tg (SNCA*A53T) 1Nbm Tg (SNCA*A53T) 2Nbm/J FVB;129S6-*Snca^tm1Nbm^* Tg (SNCA*A30P) 1Nbm Tg (SNCA*A30P) 2Nbm/J	X	X	A53T-*SNCA* or A30P-*SNCA* overexpressing *Snca* KO mice	Enteric nervous system abnormalities	[80]
M83KO mice resulting from crossing M83 line with B6; 129X1-*Snca^tm1Rosl^*/J line		X	A30P-*SNCA* overexpressing *Snca* KO mice	Dopaminergic neurodegeneration	[81]
B6.Cg-*Snca^tm1Rosl^* Tg (SNCA*A30P) 192Rwm/J	X	X	A30P-*SNCA* overexpressing *Snca* KO mice	Region-specific deficits in dopamine signaling	[82]
FVB;129-Tg (Prnp-SNCA*A53T) AAub/J	X	X	A53T-*SNCA* overexpressing mice	Dysfunctional neurotransmission Impaired synaptic plasticity	[83]
FVB;129-Tg (Prnp-SNCA*A53T) AAub/J	X	X	A53T-*SNCA* overexpressing mice	Neuronal dysfunction in the absence of aggregate formation Behavioral alterations	[84]
NTac:SD-Tg (SNCA*A53T) 268Cjli	X	X	A53T-*SNCA* overexpressing mice	Dynamic changes in striatal mGluR1 but not mGluR5	[85]
B6;C3-Tg (Prnp-SNCA*A53T) 83Vle/J (also known as:A53T α-synuclein transgenic line M83)	X	X	Brain inoculation with brain homogenates from older Tg mice or with human α-syn fibrils in Tg A53T-*SNCA* overexpressing mice	Inoculation initiates a rapidly progressive neurodegenerative α-synucleinopathy	[86]
NTac:SD-Tg (SNCA*E46K) 70CJLi	X	X	E46K-*SNCA* overexpressing rats	Enhanced vulnerability to mitochondrial impairment	[87]
Double transgenic Uchl1^tm1Dgen^:Thy1-maSN	X	X	*Uch-L1* KO + *Snca* overexpressing mice	Excess α-syn worsens disease in mice lacking Uch-L1	[88]
B6; 129×1-*Snca^tm1Rosl^*/J C57B/6jxSJL F3, Tg5093		X	*-Snca KO* mice -A53T-*SNCA* overexpressing mice	α-syn expression levels do not significantly affect proteasome function	[89]
See paper for details of the animal model(s)		X	-A53T-*SNCA* transfected in WT neurons *-Snca* KO mice	Altered fatty acid composition of dopaminergic neuron PUFAs enhance α-syn oligomerization	[90]
B6;129-*Snca^tm1Su^d Sncb^tm1.1Sud^*/J See also paper for further details		X	-Cultured neurons from *Snca* KO mice overexpressing *SNCA* -See paper for other models used	Inhibition of synaptic vesicle reclustering after endocytosis	[91]
B6;129×1-*Snca^tm1Rosl^*/J	X	X	α-syn fibrils gut-injected in *Snca* KO mice	α-syn transneuronal propagation from the gut to the brain	[92]
WT vs. KOM2		X	Glial cytoplasmic inclusions-α-syn or Lewy Bodies-α-syn injected in WT mice vs. mice that express human α-syn only in oligodendrocites (KOM2)	Cellular milieu affects pathology of α-syn	[93]

n.s.n.: Non standard nomenclature.

Recent results from genetic and pharmacological experiments in a cell model showed that knocking down *LRRK2* using a shRNA-based approach reduces oxidative stress by means of mitophagy [94]. A recent review uncovers the role of mitophagy in PD and lists the genes that may be involved in such phenomenon; many of the above-described PD-related genes may participate in mitophagy regulation (see [95] and references therein).

The number of transgenic animals related to other PD-associated genes is lower. There are, however, a significant number of studies on *PRKN, PINK1,* and *DJ-1*-related transgenic lines. Remarkably, Prkn knockout animals are resistant to the toxics usually used to produce dopaminergic neurodegeneration in wild-type rats or mice. In contrast, the Pink1 knockout showed hypersensitivity to 1-methyl-4-phenyl-1,2,3,6-tetrahydropyridine (MPTP)-induced dopaminergic neuronal loss and to activation of some of glutamate receptors (both ionotropic and metabotropic), thus affecting excitatory neurotransmission. Studies with transgenic lines also show that the *PRKN* gene product is involved in protein processing, particularly via ubiquitination. In addition, the double mutant line, so called “TwinkPark” mice showed decreased mitochondrial function and altered membrane potential.

**Table 4 ijms-22-04643-t004:** Findings in animal models related to *Prkn, Lrrk2* and *Pink1.*

Gene	Animal Model (s)	In Vivo	In Vitro	Expression Level	Main Findings	Ref.
*PRKN*	B6.129S4-*Prkn^tm1Shn^*/J	X	X	*Prkn* KO mice	Independent regulation of parkin ubiquitination and alpha-synuclein clearance	[96]
	B6.129S4-*Prkn^tm1Shn^*/J	X	X	*Prkn* KO mice	Accelerated microtubule aging in dopaminergic neurons	[97]
	B6.129S4-*Prkn^tm1Shn^*/J		X	*Prkn* KO mice	Myotubular atrophy Impaired mitochondrial function and smaller myofiber area	[98]
	B6.129S4-*Prkn^tm1Shn^*/J		X	*Prkn* KO mice	Parkin mediates the ubiquitination of VPS35 Reduced WASH complexes	[99]
	B6.129S4-*Prkn^tm1Shn^*/J	X	X	*Prkn* KO mice	ER stress and induced inflammation levels	[100]
	B6.129S4-*Prkn^tm1Shn^*/J	X	X	*Prkn* KO mice	Behavioral impairments Amplified EtOH-induced dopaminergic neurodegeneration, oxidative stress and apoptosis Dysfunction of mitochondrial autophagy	[101]
	B6.129S4-*Prkn^tm1Shn^*/J		X	*Prkn* KO mice	Parkin promotes proteasomal degradation of Synaptotagmin IV	[102]
	B6.129S4-*Prkn^tm1Shn^*/J	X	X	*Prkn* KO mice	SNPH Cargo vesicle generation not affected	[103]
	B6.129S4-*Prkn^tm1Shn^*/J	X	X	*Prkn* KO mice	Exacerbated mitochondrial dysfunction in neurons	[104]
	B6.129S4-*Prkn^tm1Shn^*/J	X	X	*Prkn* KO mice	Increased sensitivity to myocardial infarction Reduced survival after the infarction Reduced mitophagy	[105]
	B6.129S4-*Prkn^tm1Shn^*/J	X	X	*Prkn* KO mice	Parkin antagonizes the death potential of FAF1	[106]
	B6.129S4-*Prkn^tm1Shn^*/J	X	X	*Prkn* KO mice	Acutely sensitivity to oxidative stress Inability to maintain Mcl-1 levels Death of dopaminergic neurons	[107,108]
	B6.129S4-*Prkn^tm1Shn^*/J	X	X	*Prkn* KO mice	Aberrant behavioral response to dopamine replacement therapy in PD	[109]
	B6.129S4-*Prkn^tm1Shn^*/J	X	X	*Prkn* KO mice	Resisted weight gain, steatohepatitis, and insulin resistance Abolished hepatic fat uptake	[110]
	B6.129S4-*Prkn^tm1Shn^*/J	X	X	*Prkn* KO mice	Reductions in the total catecholamine release Impaired LTP and LTD Normal levels of dopamine receptors and dopamine transporters	[111]
	B6.129S4-*Prkn^tm1Shn^*/J	X	X	*Prkn* KO mice	Requiring inflammatory stimulus for nigral DA neuron loss	[112]
	B6.129S4-*Prkn^tm1Shn^*/J	X	X	*Prkn* KO mice	Reduced respiratory capacity mitochondria (in striatal cells) Delayed weight gain Lower protection against ROS	[113]
	B6.129S4-*Prkn^tm1Shn^*/J	X	X	*Prkn* KO mice	Increased extracellular dopamine concentration in the striatum Deficits in behavioral tests	[114]
	Double-mutant “TwinkPark” mice, resulting from crossing B6.129S4-*Prkn^tm1Shn^*/J line with Twinkle^dup/+^ line	X	X	*Prkn* KO (enhanced in the *substantia nigra)* mice	Decrease of mitochondrial DNA Low mitochondrial function and membrane potential Neurobehavioral deficits	[115]
	Crossing B6.129S4-*Prkn^tm1Shn^*/J line with a *Mcl-1* +/− line (n.s.n.)	X	X	*Prkn* KO + *Mcl-1* +/− mice	Dopaminergic neuron loss Motor impairments	[107]
	B6.129S4-*Prkn^tm1Shn^*/J B6.129S4-*Pink1^tm1Shn^*/J	X	X	*-Prkn* KO mice -*Pink1* KO mice	Inflammation rescued by STING-mediated action	[116]
	B6.129S4-*Prkn^tm1Shn^*/J B6.129S4-*Pink1^tm1Shn^*/J	X	X	*-Prkn* KO mice -*Pink1* KO mice	No repression of mitochondrial antigen presentation	[117]
	B6.129S4-*Prkn^tm1Shn^*/J LEH-*Pink1^tm1sage^*		X	*-Prkn* KO mice *-Pink1* KO rats	Mitophagy of damaged mitochondria in axons requires PINK1 and Parkin	[118]
	Crossing B6.129S4-*Prkn^tm1Shn^*/J line and B6.129S4-*Pink1^tm1Shn^*/J line		X	*Prkn*/*Pink1* double KO mice	Higher levels of ATP synthase Denervated neuromuscular junctions	[119]
	B6.129S4-*Prkn^tm1Shn^*/J B6.129S4-*Pink1^tm1Shn^*/J B6.Cg-*Park7^tm1Shn^*/J	X	X	*-Prkn* KO mice *-Pink1* KO mice *-Dj-1* KO mice	Aberrant striatal synaptic plasticity in rodent models of autosomal recessive PD	[120]
	Crossing DA_SYN53_ double-transgenic (tetO-SNCA*A53T) E2Cai/J line + DAT-PF-tTA) mice with B6.129S4-Prkntm1Shn/J line or with *Pink1^tm1Zhzh^* mutation line (n.s.n.)	X	X	Overexpressing human A53T-*SNCA* in DA neurons and KO for either *Prkn* or *Pink1*	Pervasive mitochondrial macroautophagy defects Dopamine neuron degeneration	[121]
	B6;129-*Pink1^tm1Aub^*/J		X	*Pink1* KO mice	Altered spontaneous EPSCs	[122]
	B6;129-*Pink1^tm1Aub^*/J		X	*Pink1* KO mice	Mitochondrial recruitment of parkin not affected	[123]
	B6;129-*Pink1^tm1Aub^*/J	X	X	*Pink1* KO mice	Progressive mitochondrial dysfunction in absence of neurodegeneration	[124]
*LRRK2*	B6.129X1(FVB)-*Lrrk2^tm1.1Cai^*/J		X	*Lrrk2* KO mice	Alterations in protein synthesis Alterations in degradation pathways	[125]
	B6.129X1(FVB)-*Lrrk2^tm1.1Cai^*/J	X	X	*Lrrk2* KO mice	LRRK2 regulates synaptogenesis and dopamine receptor activation	[126]
	B6.129X1(FVB)-*Lrrk2^tm1.1Cai^*/J		X	*Lrrk2* KO mice	LRRK2 regulates ER-Golgi export	[127]
	B6.129X1(FVB)-*Lrrk2^tm1.1Cai^*/J	X		*Lrrk2* KO mice	Neurons have more motile axonal and dendritic growth	[128]
	C57BL/6-*Lrrk2^tm1Mjfa^*/J	X	X	*Lrrk2* KO mice	LRRK2 modulates microglial phenotype and dopaminergic neurodegeneration	[129]
	C57BL/6-*Lrrk2^tm1Mjfa^*/J		X	*Lrrk2* KO mice	Stress-Related Gastrointestinal Dysmotility	[130]
	C57BL/6-*Lrrk2^tm1Mjfa^*/J		X	*Lrrk2* KO mice	LRRK2 is required for Rip2 localization to DCVs	[131]
	C57BL/6-*Lrrk2^tm1Mjfa^*/J		X	*Lrrk2* KO mice	Significant increase of ceramide levels Direct effects on GBA1	[132]
	B6;129-*Lrrk2^tm2.1Shn^*/J	X	X	*Lrrk2* KO mice	Impairment of Autophagy	[133]
	B6;129-*Lrrk2^tm2.1Shn^*/J B6;129-*Lrrk2^tm3.1Shn^*/J		X	*Lrrk2* KO mice	Impairment of protein degradation pathways Apoptotic cell death	[134]
	C57BL/6-*Lrrk2^tm1.1Mjff^*/J	X	X	*Lrrk2* KO mice	No obvious bone alteration phenotypes	[135]
	B6.Cg-Tg(Lrrk2)6Yue/J	X	X	*Lrrk2* overexpressing mice	Autophagy suppression	[136]
	B6.FVB-Tg (LRRK2) WT1Mjfa/J	X	X	*LRRK2* overexpressing mice	Behavioral hypoactivity Altered dopamine-dependent short-term plasticity	[137]
	STOCK Tg (tetO-LRRK2*G2019S) E3Cai/J		X	G2019S-*LRRK2* overexpressing mice	Perturbed homeostasis Altered neuronal morphogenesis	[138]
	B6.Cg-Tg (Lrrk2*G2019S) 2Yue/J		X	G2019S-*LRRK2* overexpressing mice	Reduction in lysosomal pH Increased expression of lysosomal ATPases	[139]
	B6.FVB-Tg (LRRK2*G2019S) 1Mjfa/J	X	X	G2019S-*LRRK2* overexpressing mice	Synapsis gain-of-function effect of the G2019S mutation	[140]
	NTac:SD-Tg (LRRK2*G2019S) 571CJLi	X	X	G2019S-*Lrrk2* overexpressing rats	Altered bone marrow myelopoiesis Peripheral myeloid cell differentiation	[141]
	NTac:SD-Tg (LRRK2*G2019S) 571CJLi	X	X	G2019S-*Lrrk2* overexpressing rats	Enhanced α-syn gene-induced neurodegeneration	[142]
	K-14Cre-positive Gba^lnl/lnl^		X	*Gba* KO mice (except in skin)	Reduced cerebral vascularization	[143]
*PINK1*	B6.129S4-*Pink1^tm1Shn^*/J		X	*Pink1* KO mice	Pink1 is not required for ubiquitination of mitochondrial proteins	[144]
	B6.129S4-*Pink1^tm1Shn^*/J	X	X	*Pink1* KO mice	Reduced motor activity Slower locomotor activity time Absence of nigrostriatal dopamine loss	[145]
	B6.129S4-*Pink1^tm1Shn^*/J		X	*Pink1* KO mice	Impaired mitochondrial trafficking Fragmented mitochondria	[146]
	B6.129S4-*Pink1^tm1Shn^*/J	X	X	*Pink1* KO mice	Hypersensitivity to MPTP-induced dopaminergic neuronal loss	[147]
	B6.129S4-*Pink1^tm1Shn^*/J		X	*Pink1* KO mice	No significant change in Ca^2+^ currents	[148]
	B6.129S4-*Pink1^tm1Shn^*/J	X	X	*Pink1* KO mice	Pathological cardiac hypertrophy Greater levels of oxidative stress Impaired mitochondrial function	[149]
	B6.129S4-*Pink1^tm1Shn^*/J	X	X	*Pink1* KO mice	Impairments of corticostriatal LTP and LTD Impaired dopamine release	[150]
	n.s.n.		X	*Pink1* KO mice	Intestinal infection triggers Parkinson’s disease-like symptoms	[151]
	Crossing B6;129-*Pink1^tm1Aub^*/J line with dOTC line	X	X	*Pink1* KO mice overexpressing *OTC* in DA neurons	Enhanced neurodegeneration in a model of mitochondrial stress	[152]
	B6.129S4-*Pink1^tm1Shn^*/J B6.129S4-*Prkn^tm1Shn^*/J		X	*-Pink1* KO mice -*Prkn* KO mice	Enhanced sensitivity to group II mGlu receptor activation	[153]
	B6.129S4-*Pink1^tm1Shn^*/J B6.129S4-*Prkn^tm1Shn^*/J		X	*-Pink1* KO mice -*Prkn* KO mice	Reduced mitochondria functions Altered mitophagy in macrophages	[154]
	FVB;129-*Pink1^tm1Aub^* Tg(Prnp-SNCA*A53T)AAub/J		X	A53T-*SNCA* overexpressing *Pink1* KO mice	Altered mitochondrial biogenesis	[155]
	FVB;129-*Pink1^tm1Aub^* Tg(Prnp-SNCA*A53T)AAub/J	X	X	A53T-*SNCA* overexpressing *Pink1* KO mice	Exacerbated synucleinopathy	[156]
	FVB;129-*Pink1^tm1Aub^* Tg(Prnp-SNCA*A53T)AAub/J	X	X	A53T-*SNCA* overexpressing *Pink1* KO mice	Potentiation of neurotoxicity	[157]
	Atad3afl/fl Mx1CrePink1 −/− mice, resulting from crossing B6.129S4-*Pink1^tm1Shn^*/J line with Atad3afl/fl Mx1Cre line	X	X	*Pink1* KO + conditional *Atad3a* KO mice	Aberrant stem-cell and progenitor homeostasis Pink1-dependent mitophagy	[158]
	B6N.129S6(Cg)-*Atp13a2^tm1Pjsch^*/J	X	X	*Atp13a2* KO mice	Autophagy impairment Reduced HDAC6 activity	[159]
	B6N.129S6(Cg)-*Atp13a2^tm1Pjsch^*/J		X	*Atp13a2* KO mice	Harmful gliosis	[160]
	B6N.129S6(Cg)-*Atp13a2^tm1Pjsch^*/J	X	X	*Atp13a2* KO mice	Neuronal ceroid lipofuscinosis Limited α-syn accumulation Sensorimotor deficits	[161]

n.s.n.: Non standard nomenclature.

A common factor in all cases is mitochondrial affectation. It should be noted that of all the genes devoted to results on transgenic animals described so far in this section, the most related to mitochondria biogenesis and function is Pink1.

Results related to *DJ-1* (*PARK7*) were quite diverse perhaps due to the fact that some rodent models were available >15 years ago. There is also a *Drosophila melanogaster* model in which expression of a mutant gene led to motor dysfunction related to oxidative stress. *DJ-1* knockout mice displayed oxidative stress and altered regulation of autophagy that may be the consequence of previous findings: mitochondrial dysfunction and increased reactive oxygen species (ROS) production and higher sensitivity to excitotoxicity. At the CNS level, the animals have less dendritic arborization and reduced number of dendritic spines in the medium spiny neurons of the striatum.

Although the *GBA* glucocerebrosidase gene is a pretty new player in PD, it has been extensively studied for its involvement in hereditary Gaucher’s disease, which is caused by glucocerebrosides in lysosomes. When assessing the neurological abnormalities of the *GBA* transgenic lines an acceleration of the onset of PD traits was observed (e.g., motor impairment, and a downregulation of some neurotrophic factors).

Results from transgenic lines affecting *Uchl1, Atp13a2, Pla2g6, SYNJ1 and FBXO7* genes are limited although they are included in Appendix A. In all cases the disturbances observed in the transgenic lines overlapped with the traits described earlier in this section. For instance, F-box protein 7 participates with Parkin in mitophagy-associated processes [162] and VPS35 participates in the transport of vesicles from mitochondria to peroxisomes [163].

## 7. α-Synuclein: Main Character or Supporting Actor?

Of course, α-syn is an important factor in PD and in any other synucleinopathy. However, evidence from gene interaction studies indicates that there is no particular link with other genes related to familial PD. It should be noted the possibility that α-syn may be related to *LRRK2*. Although no direct evidence is yet available, it is suggested that that the kinase encoded by *LRRK2*, the physiological function of which is not known, could phosphorylate α-syn and such phosphorylation could have an impact in PD etiopathology (see [164,165]). If it were the cause of the disturbances leading, for instance, to dopaminergic denervation in PD, α-syn would be the main character. Many monogenic diseases appear early in childhood, but this is not the case in early onset PD caused by α-syn mutations. Therefore, it is likely that some α-syn mutants are aggravating the deterioration of mechanisms of homeostasis maintenance that occur with aging. These mechanisms would affect vesicular traffic, protein ubiquitination and protein processing/degradation, eventually leading to production of α-syn aggregates. It is necessary to point out that in familial forms of PD, aggregates occur for both mutated forms of α-syn, when the *SNCA* gene is affected and non-mutated α-syn, when the gene affected is another one of those in Table 1. One conclusion could be that α-syn is an important actor but not the main one. At the level of PD-related genes, there does not seem to be a main character because every gene product is necessary for the essential functions occurring in the dopaminergic neurons of the substantia nigra. Any mutation of those genes would lead to a gene product that would negatively affect cell function. Such an alteration never causes the disease to appear in the first years of life, thus suggesting that there is a progressive loss of function (failure to appropriately maintain homeostasis) consistent with the progressive character of the disease. Figure 3 displays two sides of an equation in which a mutant α-syn would affect mechanisms that, in turn, would lead to α-syn aggregation, and in which a mutant *LRRK2, GBA, UCHL1, VPS35, PRKN, PINK1, ATP13A2, PLA2G6, DNAJC6, SYNJ1, DJ-1/PARK7* or *FBXO7* genes would affect mechanisms that in time would affect the processing of nonmutated α-syn, thus leading to α-syn aggregation.

In agreement with the aberrant formation of α-syn aggregates, the vesicular transport or the proteasomal degradation mechanism is defective or overexposed. Although the neurotransmitters(s) involved are not known, the association of some PD-related genes to endocytosis indicates that signal transduction may regulate the processing of α-syn. Could dopaminergic signaling be involved? This is a possibility because of the unique vulnerability of the nigral neurons in PD.

Mitochondria are linked to all of the events mentioned above, either through mitogenesis and mitochondria-derived vesicles or through the provision of energy. In fact, a substantial amount of protein processing, from synthesis to ubiquitination-directed degradation, requires energy, primarily in the form of ATP. Energy is also required for vesicle/endosome sorting and trafficking. Neurons rely primarily on the breakdown of glucose to meet energy requirements, and anaerobic glycolysis would not be sufficient to support all neuronal processes. Therefore, complete oxidation of glucose is required since ATP is obtained mainly through the Krebs’ cycle and the electron-transport chain. As shown in Figure 3, mitochondria are center stage in all events involving the PD-related gene products. In fact, there are several studies linking proteins of Table 2 to mitochondrial action. One example from data obtained in *Drosophila melanogaster* demonstrated that loss-of-function PINK1 results in motor deficits due to neurodegeneration mediated by mitochondrial dysfunction. The authors concluded: “*our genetic evidence clearly establishes that Parkin and PINK1 act in a common pathway in maintaining mitochondrial integrity and function in both muscles and dopaminergic neurons*” [41]. Similar results were more recently obtained in a cell model and in primary cells obtained from patients. Loss-of-function of PINK1 in HeLa cells leads to altered mitochondrion function and morphology. Cells from patients with mutated *PINK1* show altered mitochondrial morphology [42]. Another example is provided by [166], who showed that endoplasmic reticulum-mitochondrion interactions are regulated by *LRRK2* and by *PERK* products via protein ubiquitination pathways. It is worth mentioning the *MitoPark* mice, in which a mutated gene for a mitochondrial transcription factor in dopaminergic neurons led to the defective expression of proteins encoded by mitochondrial DNA and a phenotype consistent with parkinsonism, including motor deficits and progressive neurodegeneration [167]. Because the *MitoPark* mice exhibited deficits in the electron transport chain, it raised the question of why nigral dopaminergic neurons or their mitochondria were more vulnerable than other neurons. A differential expression of α-syn in different neuronal populations may be a reasonable hypothesis. However, in our opinion, the dopamine metabolism should be taken into account because it adds extra oxidative stress to dopaminergic neurons. Indeed, dopamine leads to the production of a quinone, aminochrome (the precursor of neuromelanin), which significantly contributes to the dark color of the substantia nigra in humans. The reaction of dopamine with molecular oxygen can lead to other quinones and free radicals, an oxidative load that alters mitochondrial function and can produce autophagy in dopaminergic neurons [168,169].

## 8. Concluding Remarks

Information available from genes related to familial PD led to a scenario that may be useful for explaining neurodegeneration in sporadic cases. Apart from a potential link between the kinase coded by LRRK2 and the phosphorylation of α-Syn, which is not yet substantiated, there is no direct relationship between the gene for this protein (α-Syn) and all the other genes displayed in Table 1. Hence, a common factor in familial and sporadic cases may be a progressive loss of efficacy in the mechanisms of vesicle traffic and protein handling, especially in degradation by the proteasome and the lysosome. The loss is quicker in familial cases as there is an excess burden due to difficulties in processing mutant forms of α-Syn or to difficulties in processing α-Syn by mutant components of the protein-processing machinery. For those processes, neuronal energy in the form of ATP is dependent of the full oxidation of glucose (i.e., involvement of mitochondrial Krebs’s cycle and electron chain transport). The higher vulnerability of dopaminergic neurons may come from the need of appropriate α-Syn processing and also from the oxidative stress produced by dopamine metabolism. It is reasonable to speculate that mitochondria copes with the supply of additional energy and with oxidative stress to the point where the demands for energy and antioxidant actions can no longer be satisfied, and cell death occurs: earlier in familial cases but later in sporadic cases. Hence, it can be speculated that any action to help mitochondria would be useful for preventing neurodegeneration. In general, to improve mitochondrial performance, coenzyme Q is useful; in fact, there is a correlation between oxidative stress and the deficit of this molecule in centenarians [170,171,172]. However, better options for improving the antioxidant machinery in the central nervous system are needed. The discovery of G protein-coupled receptors (GPCRs) in the mitochondria of neurons [173,174,175] may end up helping target those receptors to reduce oxidative burden or increase mitochondrial performance. Drugs acting on GPCRs constitute 35–45% of all approved therapeutic drugs and antagonists of receptors that are targets of neuroprotection (e.g., the adenosine A_2A_ receptor) may help. Could it be that the decreased risk of suffering from neurodegenerative diseases after consuming the adenosine-receptor antagonists caffeine (coffee and cola drinks) and theophylline (tea) [8,176,177,178,179,180,181,182,183,184,185,186,187,188,189] is due to an effect on neuronal mitochondria?

## Figures and Tables

**Figure 1 ijms-22-04643-f001:**
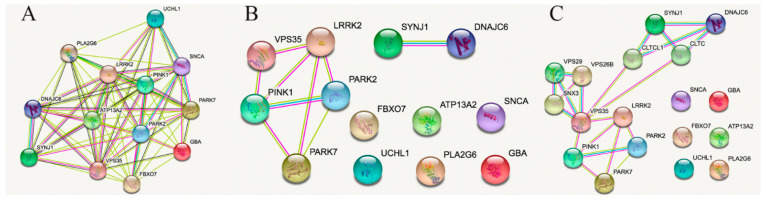
**STRING analysis of connections between (human) genes in Table 1.** Panel (**A**): using by default settings. Panel (**B**). Restricted settings (mainly leaving aside connections driven by PD-related literature; see text for details). Panel (**C**): Interaction-enriched connection pattern (see text for details). Line color code: sky blue, known interactions from curated databases; magenta, experimentally determined interactions; green, predicted from neighborhood; red, predicted from gene fusions; blue, predicted from gene co-occurrence; pastel green, textmining; black, coexpression; and clear violet, protein homology.

**Figure 2 ijms-22-04643-f002:**
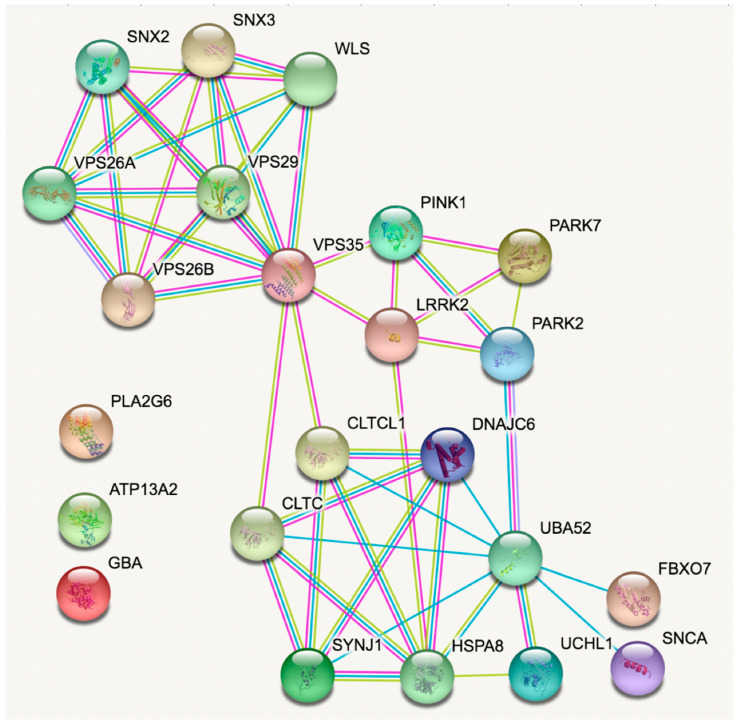
**Enriched STRING analysis of connections between (human) genes in Table 1.** Line color code; sky blue, known interactions from curated databases; magenta, experimentally determined interactions; green, predicted from neighborhood; red, predicted from gene fusions; blue, predicted from gene co-occurrence; pastel green, textmining; black, coexpression; and clear violet, protein homology.

**Figure 3 ijms-22-04643-f003:**
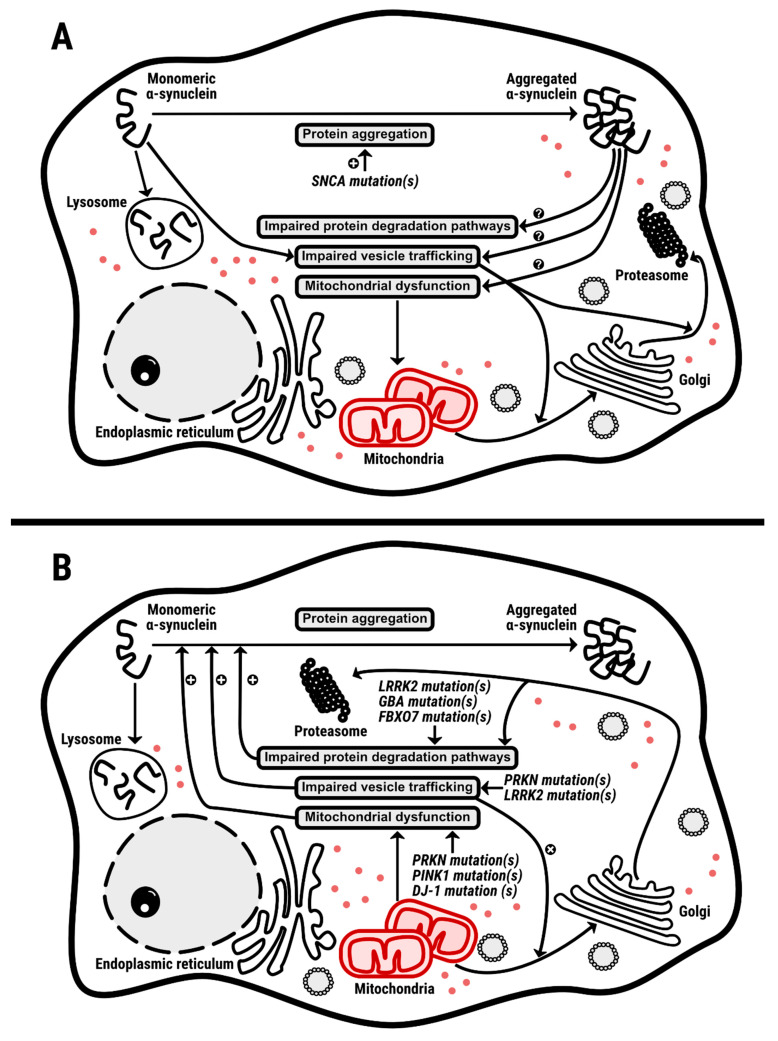
**Mechanisms of dyshomeostasis of dopaminergic neurons.** Panel (**A**). Mutations in *SNCA* leads to aggregation of mutated α-synuclein that affects the mechanisms of protein handling and vesicle transport, thus dysbalancing mitochondrial dynamics and function. Panel (**B**). Mutations in genes that affect protein processing, vesicle transport and mitochondrial function leads to aggregation of non-mutated α-synuclein. Red dots represent ATP molecules, mainly synthesized in the mitochondria of neurons, that are needed for all the processes depicted in the schemes.

**Table 1 ijms-22-04643-t001:** Genes considered in this review and having mutations likely related to early-onset PD. See description of gene products in Table 2.

*Gene*	*Inheritance*	*Proposed Disease Mechanism*	*Disease Onset*		*Mutation*	*Frequency*	*Confidence as* *Actual PD Gene*	*Year of* *Discovery*	*Animal Model*
			*Early*	*Late*					
***SNCA***	D	GoF or overexpression	Often with dementia		Missense or multiplication	Very rare	Very high	***1997,2003***	+
***LRRK2***	D	GoF	X		Missense	common	Very high	***2004***	+
***GBA***	D	Likely LoF	X		Missense or LoF	common	Very high	***2009***	+
***UCHL1***	D	LoF	NA		Missense	unclear	Low	***1998***	+
***VPS35***	D	LoF	X		Missense	Very rare	Very high	***2011***	+
***PRKN***	R	LoF	X		Missense or LoF	rare	Very high	***1998***	+
***PINK1***	R	LoF	X		Missense or loss of function	rare	Very high	***2004***	+
***ATP13A2***	R	LoF	Atypical PD		Missense or LoF	Very rare	Very high	***2006***	+
***PLA2G6***	R	LoF	X		Missense or LoF	rare	Very high	***2009***	NA
***DNAJC6***	R	LoF	X		Missense o LoF	Very rare	High	***2012***	NA
***SYNJ1***	R	LoF	(often) Atypical PD		Missense or loss of function	Very rare	Very high	***2013***	NA
***DJ-1/PARK7***	R	LoF	X		Missense	Very rare	Very High	***2003***	+
***FBXO7***	R	LoF	X		Missense	Very rare	Very High	***2008***	NA

D: Dominant; R: Recessive; NA: Not available; GoF: Gain-of-function; LoF: Loss-of-function.

**Table 2 ijms-22-04643-t002:** Description of the products of the genes in Table 1. Function retrieved from https://string-db.org/ (accessed on 10 April 2021).

*Gene*	*Ensembl Identifier*	*Protein*	*Function (Information Obtained from STRING Data Base)*
***SNCA***	ENSP00000338345	α-synuclein	Involved in regulating dopamine release and transport. Induces the fibrillization of tau protein. Reduces neuronal responsiveness to different apoptotic stimuli, thus promoting a decreased caspase-3 activation.
***LRRK2***	ENSP00000298910	Leucine-rich repeat serine/threonine-protein kinase 2	Regulates autophagy in a positive way by means of the calcium- dependent activation of the CaMKK/AMPK pathway, also involving the activation of nicotinic acid adenine dinucleotide phosphate (NAADP) receptors, increases in lysosomal pH, and release of Ca^++^ from lysosomes.
***GBA***	ENSP00000314508	Glucosylceramidase beta	Involved in the hydrolization of glucocerebroside. Localized in lysosomes.
***UCHL1***	ENSP00000284440	Ubiquitin carboxyl-terminal hydrolase isozyme L1	Deubiquitinating enzyme, generates ubiquitin monomers. Might prevent the degradation of monoubiquitin in lysosomes. Its expression is highly specific to neurons and to cells of the diffuse neuroendocrine system and their tumors.
***VPS35***	ENSP00000299138	Vacuolar protein sorting-associated protein 35	Involved in autophagy. Is part of the retromer cargo-selective complex (CSC), which is responsible for transporting select cargo proteins between vesicular structures (e.g., endosomes, lysosomes, vacuoles) and the Golgi apparatus.
***PRKN***	ENSP00000355865	E3 ubiquitin-protein ligase parkin	Ubiquitin ligase; covalently binds ubiquitin residues onto proteins. Involved in the removal of abnormally folded or damaged proteins thanks to ‘Lys-63’-linked polyubiquitination of misfolded proteins.
***PINK1***	ENSP00000364204	Serine/threonine-protein kinase PINK1	Localized in mitochondria. Protects cells against stress-induced mitochondrial dysfunction by phosphorylating mitochondrial proteins. By means of activation and translocation of PRKN participates in the clearance of damaged mitochondria via selective autophagy (mitophagy).
***ATP13A2***	ENSP00000327214	Cation-transporting ATPase 13A2	ATPase involved in the transport of divalent transition metal cations and the maintenance of neuronal integrity. It is necessary for a correct lysosomal and mitochondrial maintenance.
***PLA2G6***	ENSP00000333142	85/88 kDa calcium-independent phospholipase A2	Involved in the release of fatty acids from phospholipids. Implicated in normal phospholipid remodeling. It has also been involved in NO- or vasopressin-induced arachidonic acid release and in leukotriene and prostaglandin production.
***DNAJC6***	ENSP00000360108	Putative tyrosine-protein phosphatase auxilin	Promotes uncoating of clathrin-coated vesicles by recruiting HSPA8/HSC70 to clathrin-coated vesicles. Involved in clathrin-mediated endocytosis in neurons.
***SYNJ1***	ENSP00000409667	Synaptojanin-1	Phosphoinositide phosphatase, regulates levels of membrane phosphatidylinositol-4,5-bisphosphate (PIP2). Involved in the rearrangement of actin filaments downstream of tyrosine kinase and ASH/GRB2 by means of hydrolyzing PIP2 bound to actin regulatory proteins.
***DJ-1/ PARK7***	ENSP00000418770	Protein/nucleic acid deglycase DJ-1	Under an oxidative condition, via its chaperone activity, inhibits the aggregation of α-synuclein, thus functioning as a redox-sensitive chaperone and as a sensor for oxidative stress. Deglycates proteins and nucleotides, and the Maillard adducts formed between amino groups of proteins or nucleotides and reactive carbonyl groups of glyoxals.
***FBXO7***	ENSP00000266087	F-box only protein 7	Part of a SCF (SKP1-CUL1-F- box protein) E3 ubiquitin-protein ligase complex involved in protein ubiquitination. Role in the clearance of damaged mitochondria (mitophagy).

## Data Availability

Not applicable.

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
