# Peer review of "Genes Implicated in Familial Parkinson’s Disease Provide a Dual Picture of Nigral Dopaminergic Neurodegeneration with Mitochondria Taking Center Stage"

_ijms, 2021, doi:10.3390/ijms22094643_

Round 1
Reviewer 1 Report
Review of a manuscript “Genes implicated in familial Parkinson's disease provide a dual picture of nigral dopaminergic neurodegeneration, with mitochondria taking center stage” by Rafael Franco and coauthors submitted to IJMS.
Parkinson’s disease is a widespread neurodegenerative disease for which there is no treatment affecting the course of the disease and no reliable biomarkers for early diagnostic. The majority of Parkinson’s disease patients are sporadic, however the study of genetics of this disorder opens the molecular and cellular mechanism causing this pathology. The authors propose to investigate the interaction of genes and their products implicated in a dopaminergic denervation in Parkinson’s disease. The study of interaction of genes affecting Parkinson’s disease is important, and the authors use an innovative approach, the results if which will be interesting for the readers of the journal. The following correction should be done:
Abstract
Line 26: “Direct or functional interactions between genes or gene products are evaluated using the tools of the STRING database” the authors should explain the database. For example, “Direct or functional interactions between genes or gene products are evaluated using the tools of the database developed for protein-protein interaction networks STRING”.
Introduction
Lines 50-51: After “…why dopamine neurons of the substantia nigra die in both familial and sporadic cases, remain unknown” the authors should add a reference on the following review:” Emamzadeh FN et al.. Parkinson’s disease: Biomarkers, Treatment, and Risk Factors. Frontiers in Neuroscience, 12, 61230, August 2018, https://doi.org/10.3389/fnins.2018.00612
Lines 61-62 “…a picture of the potential events occurring in dopaminergic neurons that are degenerating”. The sentence should be rewritten as follows :” a picture of the potential events occurring in degenerating dopaminergic neurons”.
Lessons from transgenic animals
Line 174: “In accordance with the leading role of α-syn in synucleinopathies such as PD…” the authors should add here a reference on a-synuclein review: ” Synucleins and gene expression: ramblers in a crowd or cops regulating traffic?” Front. Mol. Neurosci. July 2017, 10, 1-7 224 doi: 10.3389/fnmol.2017.00224
Lines 241-242: “Surely, α-syn is an important factor in PD and in any other synucleinopathy. However, the evidence from gene interaction studies indicates that there is not any particular link with other genes related to familial PD”.
What about genes encoding enzymes phosphorylating α-synuclein?
Table 3. Findings in animal models related to the genes in table 1
This table is overloaded with information and hard to read. Moreover, the explanation of the data presented in this Table is separated from the table being located on line 170 and around, whereas the Table begins on line 304. The manuscript will benefit if the authors delete some secondary data or split the Table into two or more for easier reading.
Conclusion: The manuscript lacks Conclusion, it ends on the Table 3 data abruptly. The authors should add concluding remarks summarizing their data.
Reviewer 2 Report
This review summarizes evidence on specific gene mutations responsible for Parkinson’s disease (PD) and investigates the possible relationship among them. The authors used the Search Tool for the Retrieval of Interacting Genes/Proteins (STRING) database to assess direct or functional interactions between those genes and their products. Also, they examined transgenic animal models of PD to further speculate about possible mechanisms underlying PD. Overall, the authors support the hypothesis that changes in mitochondria function would play a key role in the neurodegeneration associated with PD.
The topic of the study is of interest. However, the manuscript is sometimes too verbose (e.g., see paragraphs 8 and 9). I have some major concerns to be addressed:
- The authors should clarify the type of the review both in the abstract and introduction (e.g., narrative review).
- The background should be improved by providing more information about neuropathological changes and consequent therapeutic approaches in PD. Also, the rationale of the review is unclear.
- Several previous review articles addressed the topic of genetic mechanisms underlying PD. The authors should clarify the novelty of their study.
- The manuscript lacks some methodological information (e.g., how many authors performed the analysis on STRING?). Also, additional information on the STRING database should be provided.
- Please, use the “full form” of the acronym “STRING” when first appearing in the text.
- I would suggest reversing the order of Table 1 and 2 by first reporting gene/proteins description and then their relationship with PD.
- Concerning Table 1, I would suggest using two separate columns to respectively report “disease onset” (e.g., early or late-onset) and main clinical features (e.g., association with dementia or other specific clinical issues such as limb dystonia in Parkin disease). Also, use more abbreviations to improve the format of the table (e.g., D for Dominant; R for Recessive; GoF for Gain of Function etc) and fill the empty cells with NA (not available) for lacking data.
- Regarding Table 2, It would be relevant to refer also to the possible antiviral role of alpha-synuclein (e.g., see the viral hypothesis of PD).
- Table 3 is too long and rich in details. I would move it in supplementary materials and use a new brief Table 3 only summarizing essential information on animal models.
- The manuscript lacks a conclusion paragraph summarizing the main findings of the review.
- Speculating about possible new therapeutic perspectives based on genetic mechanisms involved in PD occurrence would add value to the review.
- The abstract should be reorganized by providing more information on the methods and review structure. I would not anticipate the main conclusions of the review. Lines 27-35 currently sound more like a conclusion paragraph rather than an abstract.
- I would suggest making the title more appealing and brief.- The text includes some typos to correct (e.g., "invovled" in table 2).
- The manuscript currently includes only Figure S1. The other figures cited in the text (i.e., Figure 1, 2, 3) were not available at the moment of this revision.
Round 2
Reviewer 2 Report
The authors only partially addressed my comments. I believe that the contents of the review are relevant. However, the current main issue concerns the low readability of the text.
As previously indicated, the manuscript is too verbose and shows a low readability. Unfortunately, after the first authors’ revision this issue further worsened.
The abstract is still poorly organized and shows some details that could be missed (e.g., all the involved genes).
The authors did not provide the text with lacking methodological information (e.g., how many authors performed the analysis on STRING?).
Information about “disease onset” and “main clinical features” are still shown only for some genes in the same column of the Table 1.
The new conclusion section is too long and detailed.
Again, I would suggest to consider making the title more appealing and brief.